# Towards universal neural network potential for material discovery applicable to arbitrary combination of 45 elements

So Takamoto [1✉], Chikashi Shinagawa [1], Daisuke Motoki [1], Kosuke Nakago [1], Wenwen Li[1], Iori Kurata [1], Taku Watanabe[2], Yoshihiro Yayama [2], Hiroki Iriguchi[2], Yusuke Asano[2], Tasuku Onodera[2], Takafumi Ishii[2], Takao Kudo[2], Hideki Ono[2], Ryohto Sawada[1], Ryuichiro Ishitani[1], Marc Ong[1], Taiki Yamaguchi[1], Toshiki Kataoka[1], Akihide Hayashi [1], Nontawat Charoenphakdee [1] & Takeshi Ibuka [2✉]

Computational material discovery is under intense study owing to its ability to explore the vast space of chemical systems. Neural network potentials (NNPs) have been shown to be particularly effective in conducting atomistic simulations for such purposes. However, existing NNPs are generally designed for narrow target materials, making them unsuitable for broader applications in material discovery. Here we report a development of universal NNP called PreFerred Potential (PFP), which is able to handle any combination of 45 elements. Particular emphasis is placed on the datasets, which include a diverse set of virtual structures used to attain the universality. We demonstrated the applicability of PFP in selected domains: lithium diffusion in $LiFeSO_4F$, molecular adsorption in metal-organic frameworks, an order–disorder transition of Cu-Au alloys, and material discovery for a Fischer–Tropsch catalyst. They showcase the power of PFP, and this technology provides a highly useful tool for material discovery.

[1] Preferred Networks, Inc., 100-0004, 1-6-1 Otemachi, Chiyoda-ku, Tokyo, Japan. [2] Central Technical Research Laboratory, ENEOS Corporation, 231-0815, 8 Chidoricho, Naka-ku, Yokohama, Kanagawa, Japan. ✉email: takamoto@preferred.jp; ibuka.takeshi@eneos.com

Finding new and useful materials is a difficult task. Because the number of possible material combinations in the real world is astronomically large[1], methods for material exploration depending only on computer simulations are required to search through a vast number of candidate materials within a feasible amount of time.

One approach to the problem of material exploration is a quantum chemical simulation, such as a density functional theory (DFT)-based method, because many properties of materials stem from atomistic-level phenomena. However, quantum chemical calculations generally require enormous computational resources, limiting the practical use of this method in material discovery for two reasons. First, phenomena of interest in real-world applications often involve temporal and spatial scales vastly exceeding the limitations of quantum calculations, which are usually several hundreds of atoms at a sub-nanosecond scale. Second, many simulations are required to explore the configurational space during computational material discovery.

To address these challenges, several alternate computational models have been developed to directly estimate the potential energy surface of an atomic structure. For example, conventional methods called empirical potentials, which model the interaction between atoms as a combination of analytic functions, have been developed with some success, including for simple pairwise models[2], metals[3,4], covalent bonds[5], and reactive phenomena[6,7]. More recently, some machine learning-based approaches have been proposed, including Gaussian processes[8–10] and support vector machines[11].

In recent years, neural network potentials (NNPs) have rapidly gained attention owing to the high expressive power of neural networks (NNs) combined with the availability of large-scale datasets. As datasets and models evolve, the scope of NNP applications has gradually expanded. As a benchmark for molecular systems, the QM9 dataset[12,13], which covers possible patterns of small molecules, has been widely used. Initially, NNPs for organic molecules have focused on H, C, N, and O, which are the major elements in organic molecules. In subsequent studies, NNPs have been extended to include elements such as S, F, and Cl[14,15]. For NNPs targeting crystal structures[16,17], the Materials Project[18], a large-scale materials database based on DFT calculations, is often used as a benchmark dataset. The Open Catalyst Project, which targets molecular adsorption in catalytic reactions, has constructed a massive surface adsorption structure dataset known as the Open Catalyst 2020 (OC20) dataset[19,20]. In this way, the area covered by NNPs has gradually expanded.

However, significant challenges remain in the application of NNPs to computational material discovery. One unsolved issue is how to achieve the generalization needed to accurately assess the properties of unknown structures. All previously proposed datasets were generated based on known structures, and thus models trained using such datasets are only applicable to a limited configurational space. For example, the Open Catalyst Project have clearly stated that previous datasets are inappropriate for their adsorption task. By defining the system to be simulated in advance, the local configuration of atoms and combinations of elements to be generated can be reduced, thus significantly decreasing the difficulty in creating the model. However, as a disadvantage of this approach, it is necessary to recreate the NNPs and datasets for each structure of interest.

In contrast to the tasks described in previous datasets, simulations of unknown or hypothetical materials are quite common in the process of material exploration. Thus, limiting the target domain to existing materials is undesirable. This is where a major gap exists between the requirements for current NNPs and material exploration. This gap is analogous to the difference between specific object recognition and general object recognition in computer vision.

It was recently demonstrated that the NN losses in various tasks follow a power law well based on the size of the dataset and the number of NN parameters when applying a suitable model, regardless of the target domain[21,22]. Thus, NNs can achieve a high accuracy even with datasets having high diversity. This result indicated that there is a way to overcome this challenging task through the use of a sufficient dataset and architecture.

We applied the above concept to the development of an NNP. Instead of collecting realistic, known stable structures, we aggressively gathered a dataset containing unstable structures to improve the robustness and generalization ability of the model. The dataset includes structures with irregular substitutions of elements in a variety of crystal systems and molecular structures, disordered structures in which a variety of different elements exist simultaneously, and structures in which the temperature and density are varied. The NNP architecture was also designed under the premise of this highly diverse dataset. The architecture should treat many elements without a combinatorial explosion. In addition, it can utilize higher-order geometric features and handle the necessary invariances.

In this study, we created a universal NNP, called PreFerred Potential (PFP), which is capable of handling any combination of 45 elements selected from the periodic table. We conducted simulations using PFP for a variety of systems, including (i) lithium diffusion in LiFeSO$_4$F, (ii) molecular adsorption in metal-organic frameworks, (iii) a Cu–Au alloy order–disorder transition, and (iv) material discovery for a Fischer–Tropsch catalyst. All results demonstrated that PFP produces a quantitatively excellent performance. All results were reproduced using a single model in which no prior information regarding these four types of systems was applied as a prerequisite for training.

## Results

**Lithium diffusion.** The first example application is lithium diffusion in lithium-ion batteries. Lithium-ion batteries are used in various applications, such as portable electronic devices and electric vehicles. The demand for lithium-ion batteries has been increasing in recent decades, and new battery materials have been explored. One of the essential properties of lithium-ion batteries is their charge–discharge rate. Faster lithium diffusion, that is, a lower activation energy of lithium diffusion, leads to faster charge and discharge rates. DFT calculations have been widely applied to lithium-ion battery materials[23,24], and the activation energies of lithium diffusion have also been calculated for various materials[25,26]. An activation energy calculation requires accurate transition state estimations, as well as the initial and final states. The transition state is a first-order saddle point in the reaction pathway between the initial and final states. To correctly obtain the structure and energy of the transition state, a smooth and reproducible potential is required, even near the first-order saddle point, which is far from the geometrically optimized structures and harmonic vibration. The nudged elastic band (NEB) method[27] is one of the most widely used methods for obtaining the reaction path, and an improved version of this method, climbing-image NEB (CI-NEB)[28], can be used to obtain the transition state.

The tavorite-structured LiFeSO$_4$F ($P\bar{1}$) is a cathode material for lithium-ion batteries with a high voltage of 3.6 V[29]. According to existing DFT calculations, this material shows a one-dimensional diffusion, that is, the low activation energy of lithium diffusion in only a single direction[30]. We calculated the activation energy of lithium diffusion in LiFeSO$_4$F using the CI-NEB method using PFP and compared the results with those of the existing DFT calculations. It is noted that neither the crystal structure of LiFeSO$_4$F nor that of FeSO$_4$F are included in the dataset.

A delithiated structure of $LiFeSO_4F$, that is, the structure of $FeSO_4F$, is obtained by removing all lithium in the $LiFeSO_4F$ unit cell and then geometrically optimizing the cell parameters and site positions while maintaining the symmetry. All CI-NEB calculations were conducted with one lithium atom and a $2 \times 2 \times 2$ supercell of $FeSO_4F$. The chemical formula is $Li_{1/16}FeSO_4F$. The cell parameters are frozen to those of $FeSO_4F$. The diffusion paths in the [111] and [101] directions contain three diffusion hops for each, and the diffusion path in the [100] direction contains one diffusion hop[29]. There are nine NEB images for each CI-NEB calculation. PFP conducts all of this calculation on a single GPU in ~5 min.

In addition, MD simulations were performed to confirm the results of the CI-NEB calculation and demonstrate that PFP can be used for the finite-temperature dynamics simulation. The same structure as the initial state of the CI-NEB calculation was used for MD simulations. The temperature was set at 300, 325, 350, 375, and 400 K. Eight trajectories of 100 ps were generated for each temperature. The details of the MD simulation settings and the calculation method for the activation energy are described in Supplementary Note 13.

The obtained lithium diffusion paths are shown in Fig. 1, and the activation energies are shown in Table 1. The PFP qualitatively reproduces a DFT result in which $LiFeSO_4F$ exhibits one-dimensional diffusion. Furthermore, quantitatively, the PFP reproduces the DFT result with high accuracy. Although neither transition states nor reaction pathways are explicitly given in the training data for creating PFP, it is possible to correctly infer the energies of the transition states far from a stable state, as well as harmonic oscillations from such state.

**Molecular adsorption in metal-organic framework.** Metal-organic frameworks (MOFs) are a class of nanoporous crystalline materials with exceptionally high surface area. They consist of metal centers bridged by organic linkers, thereby creating diverse crystalline structures with a wide range of elements. Thus, these materials are ideal for testing the capability of PFP owing to their complex chemical structures containing organic and inorganic parts with unique crystalline pore structures. Such a system is normally difficult to reproduce using a conventional classical

interatomic potential without finetuning the potential parameters. Quantum chemical calculations, such as the DFT approach, may avoid such issues in exchange for tremendous computational costs.

To test the applicability of PFP to MOFs, some representative materials were selected, and the cell geometries were optimized. Here, it should be emphasized that none of the MOF structures are included in our training dataset; thus, this is an out-of-domain test of our model. The starting crystalline structures were obtained from the Cambridge Structure Database (CSD)[33]. The initial structures were cleaned by removing the physically adsorbed molecules in the pores of the MOFs. Water molecules that are chemically bound to the metal centers are maintained. These structures are referred to as hydrated structures. Other minor cleansing procedures were performed by adding hydrogen atoms to the framework and removing overlapping atoms to ensure physically reasonable crystal structures and stoichiometries. Dispersion interactions were also considered. The Grimmes D3 model was adopted for this purpose[34]. Notably, the dispersion correction can be calculated separately from the DFT, and adding it to PFP is still effective from a view of calculation time. To maximize the efficiency of the dispersion correction calculation, we implemented the GPU-accelerated version of DFT-D3 using PyTorch[35] and made it open-source and freely available[36]. Details of the calculation setup are provided in Supplementary Note 14.

The PFP-optimized crystal structures were compared with the experimental crystalline structures reported in the literature. Figure 2a shows the relative error in the cell volume of the MOF crystals. The individual cell parameters are provided in Supplementary Note 15. The predicted and experimental lattice parameters are in good agreement, and the mean absolute error of the cell volume is +4.5% and +3.4% with and without dispersion corrections, respectively. This translates to a deviation in the lattice parameters of approximately +0.7% for both cases. The results are encouraging because a good agreement is obtained, although MOFs are out-of-domain datasets, and no such structure is used to train the PFP.

Some MOFs have unsaturated open-metal sites that are active for the chemisorption of small molecules. For example, MOF-74 is a MOF with a one-dimensional pore structure consisting of

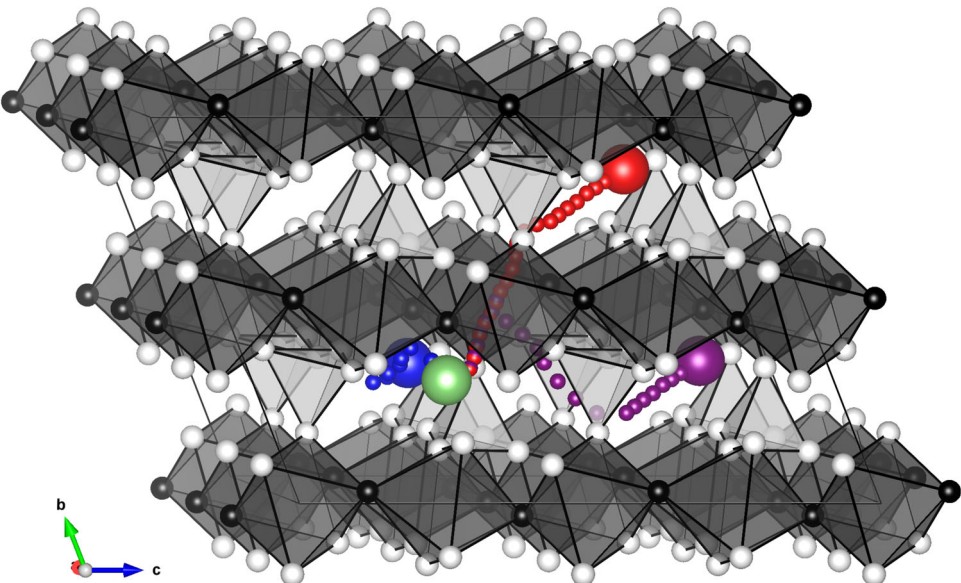

**Fig. 1 Lithium diffusion paths projected onto a 2 × 2 × 2 supercell of FeSO₄F.** Elements are represented by white spheres (oxygen), black spheres (fluorine), dark gray octahedra (iron), and light gray tetrahedra (sulfur). The small red spheres represent the lithium diffusion path in the [111] direction, from the large green sphere (initial lithium site) to the large red sphere (final lithium site). The diffusion paths in the [101] and [100] directions are represented by purple and blue spheres, respectively. The figure is drawn using the VESTA visualization package[31].

**Table 1 Activation energies for lithium diffusion through LiFeSO$_4$F at the dilution limit (i.e., through FeSO$_4$F).**

| Method | Activation energy (eV) | | |
|---|---|---|---|
| | [111] | [101] | [100] |
| DFT[30] | 0.208 | 0.700 | 0.976 |
| PFP (NEB) | 0.214 | 0.677 | 1.015 |
| PFP (MD) | 0.202 | – | – |

Note that DFT values are calculated without Hubbard $U$ corrections[32], although our datasets were calculated based on the corrections. The tests conducted by Muller et al. indicate that the corrections do not significantly affect the predicted activation energies[30].

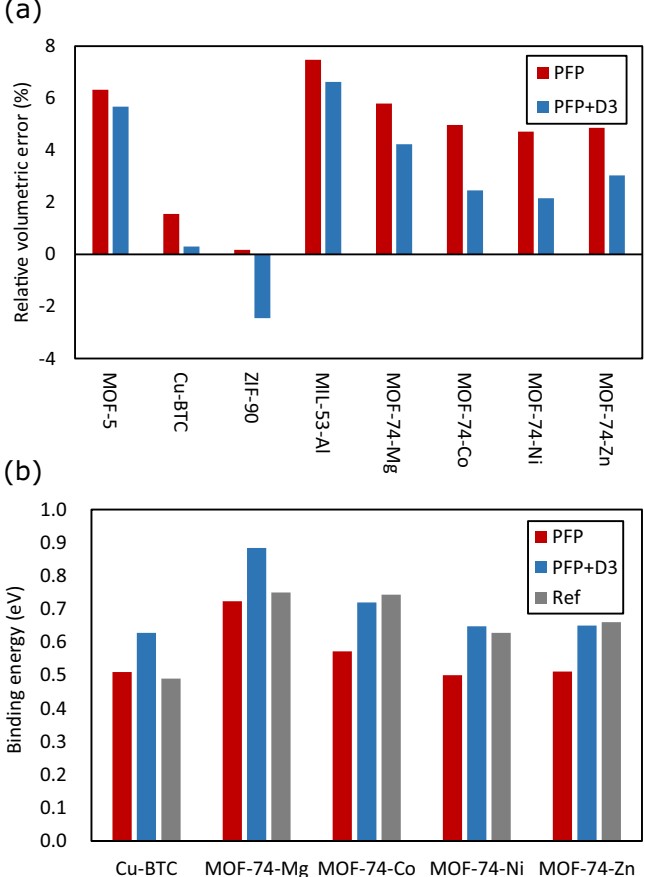

(a)

(b)

**Fig. 2 Validations of MOF structures created by PFP. a** Relative error between optimized unit cell and experimentally determined cell volumes. **b** Mean binding energies of H$_2$O molecules in selected MOFs with open metal sites with PFP, PFP+D3, and reference values. All reference values are obtained from DFT calculations.

metal(M)-oxide nodes bridged by a DOBDC ligand (DOBDC = 2,5-dioxido-1,4-benzenedicarboxylate)[37]. It is one of the early generations of MOFs, and its unique structure and properties have been well-studied[38]. There are different versions of MOF-74 with Ni, Co, Mg, and Zn, as well as of their combinations as the metals. The metal node is normally coordinated with water molecules because of the hydrothermal synthesis. The sample needs to be dehydrated by annealing at 200 °C to remove the water molecules and create open metal sites. These sites can be the locations for the adsorption of various small molecules and may act as metal centers for catalytic reactions. Another well-known example of MOFs with open metal sites is Cu-BTC (Cu$_3$(BTC)$_2$, where BTC = benzene-1,3,5-tricarboxylate)[39]. Cu-

BTC contains a copper-oxide node linked by BTC. These copper nodes can be activated by removing the chemisorbed molecules. These systems are a good test ground for the fidelity of PFP for molecular adsorption in nanoporous materials.

The mean binding energy of a water molecule is given by

$$\Delta E = -E\left(\text{MOF} + N_{\text{H}_2\text{O}} \times \text{H}_2\text{O}\right)/N_{\text{H}_2\text{O}} + E(\text{MOF})/N_{\text{H}_2\text{O}} + E\left(\text{H}_2\text{O}\right), \quad (1)$$

where $E(\text{MOF} + N_{\text{H}_2\text{O}} \times \text{H}_2\text{O})$, $E(\text{MOF})$, and $E\left(\text{H}_2\text{O}\right)$ are the total energies of the fully hydrated, dehydrated, and isolated water molecules, respectively. In addition, $N_{\text{H}_2\text{O}}$ is the number of water molecules in the system, which is 18 for all cases. Based on this definition, the more stable the compound, the more positive $\Delta E$.

Figure 2b displays the mean binding energies of water molecules in the selected MOFs with open-metal centers. The agreement between our predictions and those found in the literature is quite impressive. The largest deviation is in the case of Mg, where the error is more than 10%, whereas all other cases remain within a few percent points on average. For MOF-74 series, the agreement is better with PFP+D3. This is consistent with the fact that the literature reports use vdw-DF as the DFT functional. Conversely, in the case of Cu-BTC, the result is nearly identical to that of PFP. However, this reference uses PBE functional only, and there is no dispersion correction applied. Therefore, this is also consistent with our observation. Most importantly, PFP correctly predicts the trend in the binding energy of water molecules in a quantitative fashion.

It should be emphasized that neither the MOFs nor the metal-organic complexes examined in this section are explicitly provided in the training dataset for creating the PFP. Therefore, PFP learned to correctly predict the interaction between the metal centers and water molecules in such structures from the energies and forces of isolated molecules and periodic solids.

**Cu–Au alloy order–disorder transition.** Some precious metal alloys are well known for their catalytic activity, and extensive experimental and theoretical studies have been conducted. For example, gold–copper alloys are well-studied catalysts for the oxidation of CO and selected alcohol[40–42].

Local microscopic structures and atomic arrangements are essential for the performance of the catalyst. The Cu–Au alloy is a particularly interesting example because it is fully miscible over a wide composition range and exhibits an order-disorder transition[43]. The critical temperature is known to depend on the composition of the alloy and has been well-studied in the literature[44].

To demonstrate the applicability of PFP, we conducted Metropolis Monte Carlo (MC) simulations to investigate the transition temperature between ordered and disordered phases at various compositions of Cu–Au alloy. The calculations were applied at three different compositions: CuAu$_3$, CuAu, and Cu$_3$Au for their well-defined ordered structures. Each unit cell was expanded to $4 \times 4 \times 4$ unit cells and used as the starting geometry. The details of MC moves are shown in Supplementary Note 16.

The characterization of the resulting structures from MC simulations is summarized in Fig. 3. The computed order parameters show a clear transition from ordered to disordered phases. Perfectly ordered structures at low temperatures have well-defined order parameters and can be seen as a single point. By contrast, as the temperature increases, disturbances appear, and the plot becomes dispersed. The calculated transition temperatures are 300–400 K for CuAu$_3$, 800–900 K for CuAu, and 600–700 K for Cu$_3$Au. These trends are consistent with the

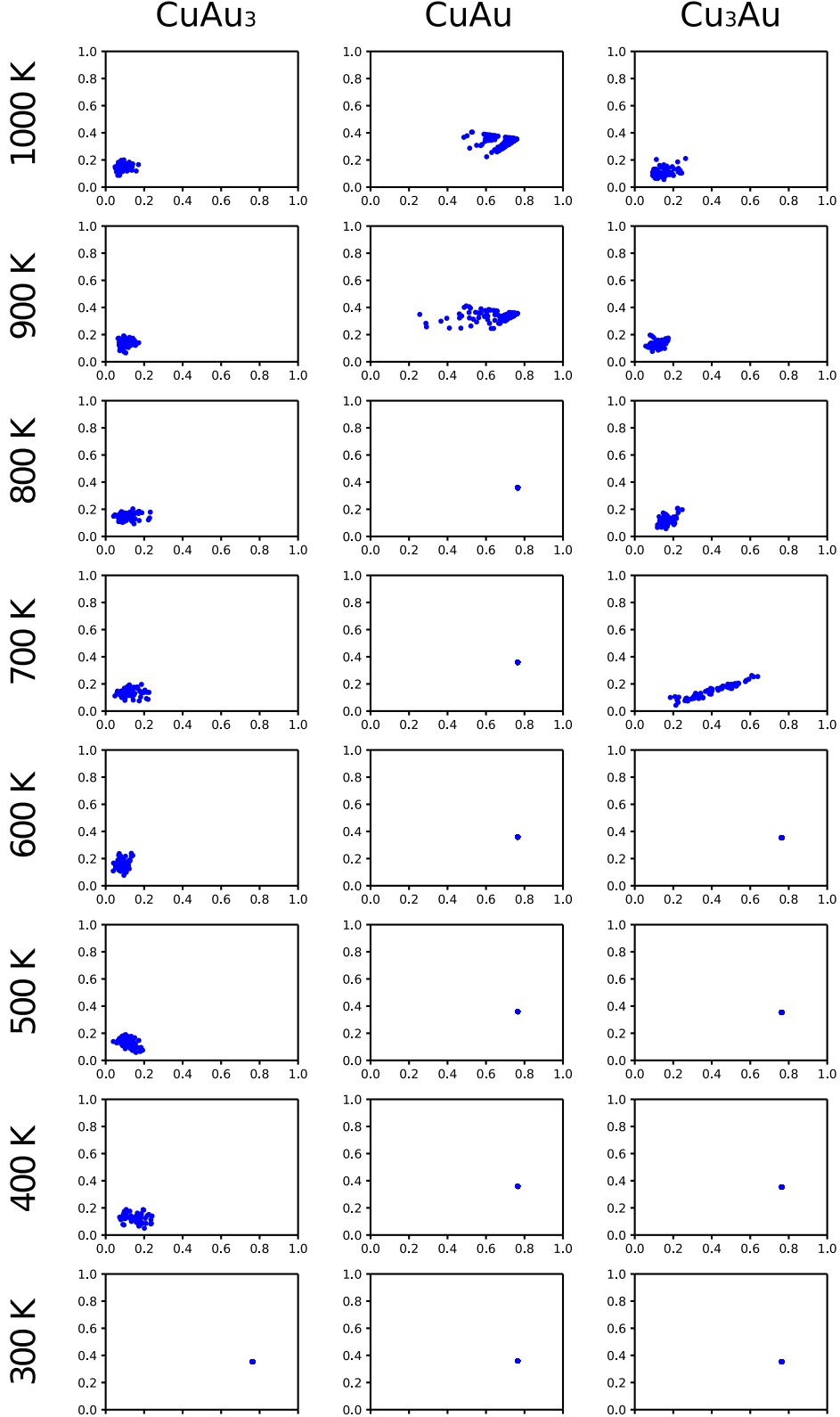

**Fig. 3 Voronoi weighted Steinhardt parameters of CuAu₃, CuAu, and Cu₃Au.** The ordinate and abscissa of each plot represent $q_4$ and $q_6$, respectively. These order parameters are calculated with respect to Cu in the case of CuAu₃ and CuAu, and with respect to Au in the case of Cu₃Au. The disordered structures can be observed as the diffused points in the figures.

reported transition temperatures (CuAu$_3$, 440–480 K; CuAu, 670–700 K; Cu$_3$Au, 660–670 K[44]) and demonstrate the applicability of PFP.

**Material discovery for a Fischer–Tropsch catalyst.** Another example of the power of PFP is given in the context of a heterogeneous catalysis. The Fischer–Tropsch (FT) reaction is a synthesis of hydrocarbons from hydrogen and carbon monoxide, involving a wide variety of elementary chemical reactions[45,46]. This reaction process is particularly important for the generation of fuel from renewable and sustainable energy sources. In this example, we focus our attention on the methanation reactions and CO dissociation processes on Co surfaces.

The methanation reactions of synthesis gases are well documented in the literature[47]. In particular, 20 elementary reactions on the Co(0001) surface have been examined, and corresponding activation energies are compared with the values reported in the literature.

Each simulation cell geometry consisted of 45 Co atoms with 5 atomic layers. Only the bottom three layers were constrained, and the rest were allowed to relax. The vacuum size was set to 10 Å ($1 Å = 10^{-10}$ m). The geometry is optimized until the maximum force of all atoms reaches below 0.05 eV/Å. The activation energy was determined by CI-NEB using 14 images for each process. Zero-point energy corrections were also included in the calculations.

Figure 4 shows a comparison of the computed activation energies between PFP and the reported values[47]. The correlation coefficient is 0.98, and the mean absolute error is 0.097 eV, indicating the high fidelity of PFP for the prediction of activation energies in this class of chemical reactions.

Backed with the high fidelity of PFP, we explored possible promoter elements for the CO dissociation reaction on a Co surface. CO dissociation is a critical part of the overall reaction mechanism of the FT process. Although it was reported to be approximately 1 eV for the activation energy of pure Co surfaces, a reduction of the activation barrier is desired, and several efforts have been reported in the literature[48]. However, DFT calculations for such exploration demand a high computational cost, and PFP can accelerate such a screening process. Specifically, we explored the CO dissociation reaction pathways by CI-NEB on the Co(11$\bar{2}$1) step surface. In the promoter search process, a Co atom was randomly replaced with a promoter element, and the CI-NEB calculations were repeated over the surface. The CI-NEB was repeated 10 times on each surface, and a list of activation energies was obtained.

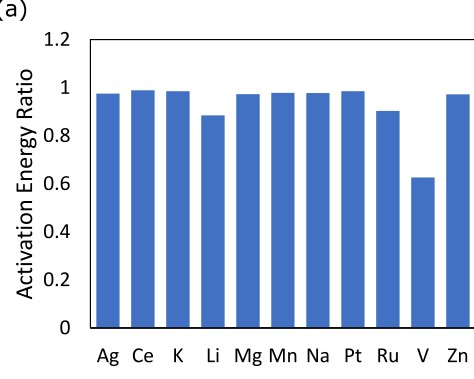

(a)

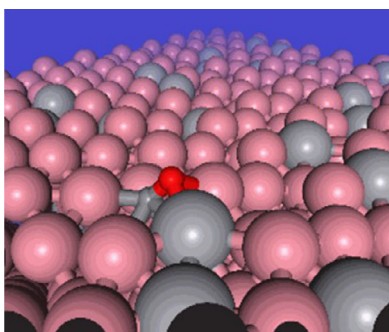

(b)

**Fig. 5 CO molecule interactions on the Co(11$\bar{2}$1) surface. a** Normalized activation energies of CO dissociation. **b** CO adsorbed configuration of a Co(11$\bar{2}$1) surface with V promoters. The representative atoms are Co (pink), O (red), C (small gray), and V (big gray).

Because they are often found in the literature as promoters of certain reactions, we chose the following 11 elements (Ag, Ce, K, Li, Mg, Mn, Na, Pt, Ru, V, and Zn) for our study. The results are summarized in Fig. 5a. Among the list, the most significant reduction (~40%) was found with V, whereas the others showed a minor impact on the activation barrier. The lowest energy configuration of CO adsorbed Co(11$\bar{2}$1) with V is shown in Fig. 5b. The CO molecule was found to lie across the Co and V bridge sites. In fact, some experimental studies have already reported a significant reduction in the activation energy of Co by V, although we identified the element without any prior knowledge from the literature[49,50]. The agreement between our findings and the literature is consistent. It is encouraging to note that our approach can facilitate the use of PFP in complex systems such as a heterogeneous catalysis.

## Discussion

We developed a universal NNP called PFP, which operates on systems with any combination of 45 elements.

The results indicate that a single NNP model can describe a diverse set of phenomena with high quantitative accuracy and low computational cost. In addition, it was also shown that PFP can reproduce structures and energetic properties that were not envisioned during the design phase. The detailed correspondence between the results and the PFP dataset is shown in Supplementary Note 11. The reproduction of the simulations in the Results section using OC20 DimeNet++ model is shown in Supplementary Note 12. Our results suggest that the approach to constructing a unified NNP, instead of training an independent NNP for each target task, is promising. Further comparison of calculation time between PFP and DFT is included in Supplementary Note 6. Although DFT calculations or other electronic structure calculations from first principles are still considered to

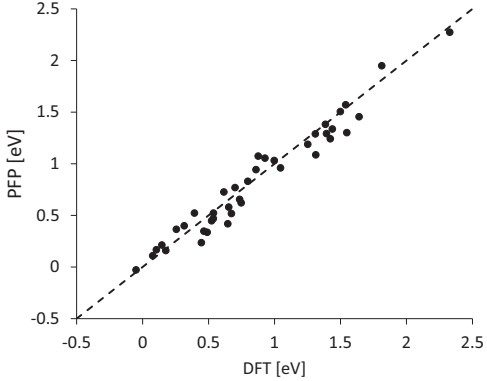

**Fig. 4 Comparison of the activation energies of methanation reactions of synthesis gas on Co(0001).** The ordinate and abscissa represent the PFP prediction and reference DFT values, respectively. The zero-point energy corrections of the transition states are also included in the data.

be reliable because of the strong physics background, PFP can greatly mitigate another limitation of atomistic simulations caused by the time and space scales. The combined study of DFT and PFP or experiments using PFP-based screening will also accelerate the field of material discovery. The simulation script files and output data are provided in Supplementary Data 1.

The result of the Fischer–Tropsch catalyst is an example of applying PFP to an actual material discovery task. This is a typical case in which NNP is able to achieve the following three properties at the same time: (1) the ability to handle a wide variety of elements, (2) the ability to handle phenomena that were not assumed at the time of training, and (3) a significantly faster speed than that of DFT.

These results further confirm that PFP is versatile and applicable for screening a wide range of materials without prior knowledge of the atomic structures in the target domain.

## Methods

**Dataset systems and structures**. In this study, we generated an original dataset which covers various systems. See Supplementary Note 10 for the definition of each subcomponents and the detailed calculation conditions on how to generate them, and Supplementary Note 19 for the statistical information of our dataset. The summary of our dataset is shown below. The visualized examples of typical structures are shown in Supplementary Note 7.

Early examples of large datasets with quantum chemical calculations include QM9[12,13] and the Materials Project[18]. They were generated by conducting DFT calculations on various molecules or inorganic materials and collecting physical properties in geometrically optimized structures to accelerate drug or material discovery. Although they have been utilized for predicting physical properties such as HOMO–LUMO gaps or formation energies of optimized structures, they are insufficient for generating universal potentials for new material discovery because they mainly focus on optimized structures. In particular, the reaction, diffusion, and phase transitions are dominated by structures far from the optimized structures. By contrast, it is unsuitable to sample geometrically random structures. Because the probability distribution of the structures follows a Boltzmann distribution, geometrically random structures that tend to show much higher energies compared to optimized structures rarely appear in reality. Therefore, it is important to cover as many diverse structures as possible while limiting those showing valid energies.

To achieve this, ANI−1[14], ANI−2x[15], and tensor-mol[51] sampled not only geometrically optimized structures of various molecules, but also their surrounding regions using NMS, MD, or meta-dynamics. Using these methods, we can obtain datasets to generate the potential to reproduce phenomena with large structural deformations, such as protein–drug docking, which is important in drug discovery. However, these datasets focus only on molecules and do not cover systems such as crystals and surfaces. One recent study that deserves attention is OC20[20], which has an order of magnitude larger number of data than previous studies. Nevertheless, this dataset also focuses on catalytic reactions and only contains data on the adsorbed structures. As we have shown, it is worth noting that these adsorbed structures are generated with known stable structures. As a result, the accuracy of the energy predictions is much lower for structures that depart from known stable structures.

Following these insights and issues, we generate an original dataset that covers all systems with molecular, crystal, slab, cluster, adsorption, and disordered structures, as shown in Table 2. For each system, we sampled various structures, such as geometrically optimized structures, vibration structures, and MD snapshots, to collect the data necessary to obtain a universal potential.

Our dataset consists of a molecular dataset calculated without periodic boundary conditions, and a crystal dataset calculated with periodic boundary conditions. Each dataset contains the structure and corresponding total energies and forces obtained through DFT calculations. The crystal dataset also includes the atomic charges. The molecular dataset supports nine elements: H, C, N, O, P, S, F, Cl, and Br. There is maximum of eight atoms from among C, N, O, P, and S in a molecule. In addition to stable molecules, unstable molecules and radicals are also included. Various structures are generated for a single molecule through geometrical optimization, NMS, and MD at high temperatures. The two-body potentials for almost all combinations of up to H–Kr are also calculated as additional data. For the crystal dataset, 45 elements are supported, as shown in Fig. 6. This includes a variety of systems, such as bulk, cluster, slab (surface), and adsorption on slabs. Non-stable structures, such as Si with simple cubic ($Pm3m$) or FCC ($Fm3m$) structures or NaCl with a zincblende structure ($F\bar{4}3m$), as well as non-optimized structures, are also included in the crystal dataset. For the bulk, cluster, and slab, we generated structures by changing the cell volumes or shapes, or by randomly displacing the atomic positions, instead of applying the NMS method. For the adsorbed systems, we generated structures with randomly placed molecules in addition to the structure-optimized ones using PFP. Disordered structures are generated using MD at high temperatures for randomly selected and placed atoms.

Molecules are also included in the crystal dataset. The two-body potentials for almost all combinations of up to H–Bi were also calculated. The computational resources used to acquire these datasets were $\sim 6 \times 10^4$ GPU days.

We provide an atomic structure dataset called the high-temperature multi-element 2021 (HME21) dataset, which consists of a portion of the PFP dataset[52]. See Supplementary Note 17 for further details.

**Training with multiple datasets**. In addition to the above molecular and crystal datasets, we used the OC20 dataset as a training dataset. This means that there are multiple datasets generated by different DFT conditions that are inconsistent with each other. Attempting to merge these datasets simply does not yield a good performance in practice. Overlapping dataset regions with different DFT conditions may have harmed the training because each data point would have resulted in inconsistent energy surfaces.

However, because these datasets are well sampled in each area of strength, it is desirable to use as much data as possible to improve the generalization. Therefore, we assigned labels corresponding to the DFT conditions during training and trained the entire dataset concurrently. During inference, it is also possible to select which DFT condition to infer by assigning labels in the same way as during training. This approach makes it possible to learn multiple mutually contradictory datasets with high accuracy. In addition, as the model learns the consistent properties of all datasets and the differences in each, it is expected that domains that have only been computed in one DFT condition will be transferred to the inference under other DFT conditions. The additional benchmark is shown in Supplementary Note 20.

Considering that datasets will become even larger in the future, the mechanism for the simultaneous training of datasets with different DFT conditions will become more important.

We considered the crystal dataset as the most basic one. All applications shown in this study are calculated in the corresponding calculation mode.

**DFT calculation conditions**. DFT calculations for the molecular dataset are carried out using the $\omega$B97X-D exchange-correlation functional[61] and the 6-31G(d) basis set[62] implemented in Gaussian 16[63]. To reproduce the symmetry-breaking phenomena of the wavefunction, such as a hydrogen dissociation, we carry out unrestricted DFT calculations with a symmetry-broken initial guess for the wavefunction. However, for geometrical optimization calculations, we carry out restricted DFT calculations. We only consider singlet or doublet spin configurations except for diatomic potentials.

Spin-polarized DFT calculations for the crystal dataset are carried out using the Perdew–Burke–Ernzerhof (PBE) exchange-correlation functional[64] implemented in the Vienna Ab-initio Simulation Package[65–68] (VASP), version 5.4.4, with GPU acceleration[69,70]. The projector-augmented wave (PAW) method[71,72] and plane-wave basis are used with a kinetic energy cutoff of 520 eV and pseudopotentials, as shown in Fig. 6. Here, $k$-point meshes are constructed based on the cell parameters and the $k$-point density of 1000 $k$-points per reciprocal atom. However, Γ-point-only calculations are carried out for structures with vacuum regions in all directions, such as molecules and clusters. For the DFT calculations on a wide variety of systems, including insulators, semiconductors, and metals, under the same conditions, we use Gaussian smearing with a smearing width of 0.05 eV. The generalized gradient approximation with Hubbard $U$ corrections (GGA+$U$) proposed by Dudarev et al.[32] is used with the $U−J$ parameters shown in Table 3. To maintain the consistency of the energies and forces in the different systems, we use the GGA+$U$ method for all structures, including metallic systems. To consider both ferromagnetism and anti-ferromagnetism, we carry out a calculation with both parallel and anti-parallel initial magnetic moments and adopt the result with the lowest energy. Nevertheless, for some systems, we carry out the calculations using only parallel initial magnetic moments. Bader charge analyses[73–76] are carried out to obtain atomic charges.

**Trained properties**. The energy of the system, atomic forces, and atomic charges are used for the training procedure. Atomic charges are considered as supplementary data. Although they are neither directly used to calculate energy nor to simulate the dynamics, they are expected to have information about the local environment of the atoms.

**Neural network architecture**. The TeaNet[60] architecture was used for the base NNP architecture of the PFP. The TeaNet architecture incorporates a second-order Euclidean tensor into the GNN and performs message passing of scalar, vector, and tensor values to represent higher-order geometric features while maintaining the necessary equivariances. For a detailed explanation of the TeaNet architecture, such as step-by-step operation, the method of treating invariances, schematic comparison between the other models, and the reported original performances for both learning procedure and MD applications, see the original material[60]. The benchmark score using HME21 dataset is shown in Supplementary Note 18. The corresponding code is provided in Supplementary Data 2. In addition, OC20 dataset benchmark is shown in Supplementary Note 1. The comparison of NNP architectures from the view of invariances are shown in Supplementary Note 9.

**Table 2 Comparison of DFT calculated datasets that can be used to train the neural network potential.**

| Dataset | Systems | | | | | | Structures | | | | # of | |
|---|---|---|---|---|---|---|---|---|---|---|---|---|
| | Molecule | Bulk | Cluster | Slab | Adsorption | Disorder | Opt. | Vib. | MD | TS | Elements | Data |
| Materials Project[18] | | ✓ | | ✓ | | | ✓ | | | | Unlimited | $>1 \times 10^5$ |
| OQMD[53] | | ✓ | | | | | ✓ | | | | Unlimited | $8 \times 10^5$ |
| NOMAD[54] | | ✓ | | | | | ✓ | | | | Unlimited | $>5 \times 10^7$ |
| Jarvis-DFT[55] | | ✓ | | | | | ✓ | | | | Unlimited | $>4 \times 10^5$ |
| AFLOW[56] | | ✓ | ✓ | | | | ✓ | | | | Unlimited | $>3 \times 10^6 (*1)$ |
| QM9[12,13] | ✓ | | | | | | ✓ | | | | 5 | $1 \times 10^5$ |
| PubChemQC[57] | ✓ | | | | | | ✓ | | | | 30 | $>3 \times 10^6 (*2)$ |
| MD17[58] | ✓ | | | | | | | ✓ | | | 4 | $9 \times 10^6$ |
| $S_N2$ reactions[59] | ✓ | | | | | | ✓ | ✓ | | ✓ | 6 | $4 \times 10^5$ |
| ANI-1[14] | ✓ | | | | | | ✓ | ✓ | ✓ | | 5 | $2 \times 10^7$ |
| ANI-2x[15] | ✓ | | | | | | ✓ | ✓ | ✓ | | 7 | $9 \times 10^6$ |
| COMP6v2[15] | ✓ | | | | | | ✓ | ✓ | ✓ | | 7 | $2 \times 10^5$ |
| tensor-mol 0.1 water[51] | ✓ | | | | | | | | ✓ | | 2 | $4 \times 10^5$ |
| tensor-mol 0.1 spider[51] | ✓ | | | | | | | | ✓ | | 4 | $3 \times 10^6$ |
| TeaNet[60] | ✓ | | | | ✓ | | | | ✓ | | 18 | $3 \times 10^5$ |
| OC20[19,20] | | | | ✓ | | | ✓ | ✓ | ✓ | | 56(*3) | $1 \times 10^8$ |
| PFP molecular dataset (ours) | ✓ | | | | | | ✓ | ✓ | ✓ | | 9 | $6 \times 10^6$ |
| PFP crystal dataset (ours) | ✓ | ✓ | ✓ | ✓ | ✓ | ✓ | ✓ | ✓ | ✓ | | 45 | $3 \times 10^6$ |

(*1): The number is checked on May 24, 2021. (*2): The number is taken from [57], and is updated weekly. (*3): The number was checked using only the training dataset of version 1.

**Fig. 6 The 45 elements supported by PFP are colored in the periodic table.** Pseudopotentials used in the DFT calculations for the PFP crystal dataset are also shown in the periodic table. These are supplied with the VASP package, version 5.4.4, and chosen by the Materials Project[18].

**Table 3 List of $U-J$ parameters. Values except for Cu are used in the Materials Project[18], and the value for Cu is determined by Weng et al. [77].**

| Elements | V | Cr | Mn | Fe | Co | Ni | Cu | Mo | W |
|---|---|---|---|---|---|---|---|---|---|
| U–J (eV) | 3.25 | 3.7 | 3.9 | 5.3 | 3.32 | 6.2 | 4.0 | 4.38 | 6.2 |

To adopt the PFP dataset, several architectural modifications were made in this study. The major modifications are shown below.

First, the Morse-style two-body potential term is introduced in addition to the TeaNet architecture. The main purpose is to reproduce the short-range repulsion effect. When the distance between two atoms becomes much closer than the stable bond distance, the nuclear repulsion force becomes dominant, and the energy increases rapidly as the distance decreases. Usually, these types of structures are not observed during the dynamics simulations. In addition, the requirement of accurate energy estimations is not considered for these high-energy structures. However, if the NNP does not learn these structures, it is difficult to reproduce the above nature when the structure accidentally contains a very close atom pair. It is possible to estimate extremely low energies for these structures. As an example of the application of PFP, such a scenario may be fatal when performing exploratory atomic system calculations, such as structure sampling using Monte Carlo methods or structure estimation using generative models. From the aspect of the training procedure, extremely large values make the training process difficult. Therefore, we trained the parameters of the Morse-style two-body potential for all possible combinations of elements independently and added them to the energy term separately. This modification is aimed to expand the practical convenience. Neither the dataset nor the applications presented in this study deal with such an energetically extreme region, and it is assumed that the introduction of the two-body potential has negligible effect.

As described in the "Dataset systems and structures" section, the dataset contains multiple DFT conditions, such as different basis functions or exchange-correlation functionals. The data points are consistent at a high accuracy level under the same computational conditions but not between different computational conditions. This difference cannot be eliminated by zero-point shifts or linear multiplications of the energy. Unifying these sub-datasets directly is considered to provide unintended virtual energy gaps. To address this problem, the DFT condition is set as an additional input label during the training. Label information is also needed during inference. This is referred to as the calculation mode of PFP. Therefore, the calculation mode has two aspects. One is to enable the training of multiple datasets that have different conditions simultaneously, and the other is to provide a feature to select those conditions for users.

The output of the TeaNet architecture is modified to output atomic charges in addition to the total energy. Charges are considered auxiliary values. Unlike the

charge equilibrium method, the charges are calculated using the forward path of the GNN. The explicit Coulombic interaction term was not included. This modification has two purposes. One is to allow PFP users to use the output charges for post-processing molecular dynamics. The other is to increase the number of learned properties for the same DFT calculations.

**NNP characteristics**. In this section, the characteristics of PFP are summarized from the perspective of NNP architecture.

PFP, or its GNN architecture TeaNet, has invariance for E(3) transformations. In other words, PFP holds rotational invariance, translational invariance, and mirror-image reversal invariance. In addition, PFP is a fully local interaction model. This means that the information of the local structure cannot propagate over an infinite distance. For example, suppose there are two molecules, A and B, that are sufficiently far apart. It is guaranteed that whatever state molecule B is in (i.e., stationary, in the middle of a chemical reaction, or artificially erased at a certain moment during the simulation), molecule A is, in principle, unaffected. The number of GNN layers is 5. The cutoff distance of the GNN layer depends on the stage of the layer; they are set to 3, 3, 4, 6, and 6 Å, respectively. This was determined by considering the balance between computational cost and accuracy. This can be regarded as a special case where all cutoff distances are equal to 6 Å, which is the original TeaNet architecture. Since GNN is multi-layered, the information of the atoms propagates through the network to their neighbors, and thus the distance at which one atom interacts with another is the summation of those cutoff distances, which is 22 Å. The physical counterpart of this phenomenon is the long-range interactions that occur as a result of the connected electron orbitals, such as metallic bonds and interactions through $\pi$-bonds.

Those properties are beneficial for improving generalization. Since both invariances and the local interaction model are satisfied, the spatial invariances are maintained for two spatially separated molecules independently. Furthermore, the extensive energy properties are preserved. In other words, when a system is composed of the sum of separated subsystems, the energy is also the sum of such subsystems. In addition, when the size of the system is doubled in the direction of the periodic boundary, the energy of the system is guaranteed to double.

PFP follows TeaNet's differentiable nature up to a higher order with respect to the position of the atom. The smoothness of the energy surface is a property directly related to the stability of the calculation, both in minimization calculations, such as structural relaxation calculations and NEB methods, and in long-time dynamics calculations. Furthermore, although molecular dynamics simulations use forces corresponding to first-order derivatives of energy, they often require quantities corresponding to higher-order derivatives, such as elastic modulus calculation, or minimization based on the quasi-Newton method.

The additional benchmark of the PFP architecture using OC20 dataset is shown in Supplementary Notes 1, 3, 4 and 8. The regression score for our dataset is shown in Supplementary Note 2 and Note 5.

## Data availability

The simulation script files and output data corresponding to the result section data generated in this study are provided in Supplementary Data 1. The atomic structure dataset called the high-temperature multi-element 2021 (HME21) dataset generated in this study have been deposited in open access repository figshare under accession code https://doi.org/10.6084/m9.figshare.19658538[52].

## Code availability

The code for NNP architecture benchmark using HME21 including TeaNet (base model of PFP) implementation with the trained parameters is provided in Supplementary Data 2. PFP is provided in the proprietary software named Matlantis. The code and trained parameters are not open-source, but PFP can be used to reproduce the results through software-as-a-service (https://matlantis.com/).

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

## Author contributions

S.T. developed the PFP model. S.T., C.S., I.K., R.S., and M.O. collected the dataset. D.M. started this project and conducted initial research. K.N. conducted the survey and developed torch-dftd. W.L., I.K., and H.I. benchmarked PFP accuracy. C.S., T.W., and Y.Y. created result cases. S.T., C.S., T.W., and Y.Y. wrote the manuscript. W.L., T.W., R.I., and M.O. elaborated the manuscript. H.I., Y.A., T.O., T.Ish., T.Ku., and H.O. validated PFP in various material fields. T.W., Y.Y., Y.A., T.O., T.Ish., R.S., R.I., M.O., T.Y., T.Ka., and A.H. developed simulation methods using PFP. N.C. took the benchmark. K.N., T.Ku., and H.O. managed the team. T.Ib. supervised this project.

## Competing interests

S.T., C.S., D.M., K.N., W.L., I.K., R.S., R.I., O.M., T.Y., T.Ka., A.H. and N.C. are employees of Preferred Networks, Inc. T.W., Y.Y., H.I., Y.A., T.O., T.Ish., T.Ku., H.O., and T.Ib. are employees of ENEOS Corporation. In addition, Preferred Computational Chemistry, Inc. provides software-as-a-service named Matlantis which is based on the product of this research.
