## [Peer Review File · Nature Communications]

REVIEWER COMMENTS

Reviewer #1 (Remarks to the Author):

This manuscript reports a machine learning potential trained on both molecular and crystal data with broad coverage across the periodic table, and in principle able to simulate any combination of 45 distinct elements.

This would be a big achievement and definitely, the sort of compelling article that warrants publication. However, the information presented in the paper is very incomplete, to the point that it is hard to evaluate what is actually new (the model? the data?) the benchmarks are all one-off and there is no comparisons with other approaches and the traceability and reproducibility considerations are appalling (No meaningful data availability statements, no code availability statement, no description of the ML model or hyperparameters, no description of the training data or the train/test splits)

There are some language issues (some examples below) that may have hindered my assessment, but my criticism goes well beyond that.

Unfortunately, in this state, a manuscript cannot be considered for publication. The authors would need to conduct a deep revision with a focus on rigorous evaluation of how their model works compared with other baselines, where the innovations and improvements lie, and a lot more focus on what the ML model and the training data are. No one could even attempt to reproduce any of what is reported in this manuscript, with the given information.

Detailed comments:

Data

"we generate an original dataset that covers all systems" So what exactly is in there? What chemistries? In what numbers? This should be very clear and very up front in the main paper. If the dataset is new, it's a result, not a method, and should be addressed very explicitly and rigorously.

"we sampled various structures, such as geometrically optimized structures, vibration structures, and MD snapshots," So which was applied to each structure and in which numbers?

"Various structures are generated for a single molecule through geometrical optimization, NMS, and MD at high temperatures." What temperatures? How was NMS done? There is nothing here that allows reproducibility

"we generated structures by changing the cell volumes or shapes, or by randomly displacing the atomic position" By how much? Which random distribution? Uniform? Gaussian? With which parameters?

"In addition to the above molecular and crystal datasets, we used the OC20 dataset as a training dataset." Then why is the dataset size smaller than OC20? The whole should be larger than the sum of the parts.

"The computational resources used to acquire these datasets were approximately 6×10^4 GPU days" what is meant to acquire the datasets? Were they recalculated? Calculated de novo? What about the CPU time?

"During inference, it is also possible to select which DFT condition to infer by assigning labels in the same way as during training" So which conditions to infer were selected in all the experiments in the paper? Out of which ones?

It is not clear if this paper has innovated a bit, a lot, or nothing at all on the dataset.

Will the dataset be released? Made available upon request? Kept private? The authors seem to praise the open datasets made by others in the introduction (and leverage them in their own work?)

Models

Table VI is too detailed to be in the main text and an unnecessary distraction. This paper is not comparing against any methods, so why review them? On the other hand, it is not clear what "ours" actually is.

I seem to understand it is the same as TeaNet? Then the authors should be very explicit about if. If it is, the architecture needs to be described layer by layer, interaction block by interaction block. It is not clear if this paper has innovated a bit, a lot, or nothing at all on the interatomic potential part.

Experiments:

Despite all the discussion of dynamics. There are no examples of using the potential to dynamics.

Was LiFeSO₄F in the training data? Which geometries for it? The agreement in Table I is outstanding. (DFT itself is barely reproducible to that degree of accuracy and NEB calculations are notoriously fickle to initial parameters or number of images. The agreement is even more surprising due to the difference in the use of U corrections!!)

"Although neither transition states nor reaction pathways are explicitly given in the training data for creating PFP, " But were MD or NMS or randomly-distorted LiFeSO₄F geometries in the training data?

Where does the initial structure of LiFeSO₄F come from? Was it relaxed with PFP?

Why not test NN-MD for LiFeSO₄F ? With such low barrier hopping should be easy to see, at least in one dimension.

Similar questions arise about the MOF. Are there MOFs in the training data. Is this MOF in the training data. Where does the initial geometry to relax with PFP come from?

The argument Table II makes is confusing. Often the cell changes less upon hydration than the error w.r.t experiment. Again, how the cells were initialized and whether this a test in the training or in the test domain is of vital importance to evaluate what PFP is bringing to the table.

Does Table III recover the rank ? Is Literature experimental or theoretical values?

Does the CuAu experiment use geometrical information? "An arbitrary pair of atoms are swapped, and the energy change (ΔE) is recorded" Were the structures relaxed after the swap? Or kept in the parent lattice parameter?

With so much content (and the need to add so much more) The details of how Metropolis Hastings work can be deferred to the Methods or SI

How does the reader infer a transition temperature from Figure 3?

The agreement with experiment does not seem great and seems to go range around 100 to 200 K. That is worse than a typical cluster expansion

"In this example, we focus our attention on the methanation reactions and CO dissociation processes on Co surfaces." Was this preset in the training data? Is this from OC20?

Were the V experiments confirmed with DFT ?

Last but not least. There is no comparison with any baselines whatsoever. Other NN architectures? And/or training over only a datasets?

Language:

"Therefore, PFP possesses the advantages of both universality and a low computational cost" This is clearly not a conclusion of the preceding text

Reviewer #2 (Remarks to the Author):

In this manuscript, the authors report a universal neural network potential that they claim can be applied to an arbitrary combination of 45 elements. They attempt to solve a key problem with current machine learning based potentials – generalization to new material structures. They created two datasets, PFP molecule and PFP crystal, and developed a GNN based neural network to train a neural network potential. They demonstrate the generalization capability of their model by applying it to 4 very different problems related to material discovery: 1) computing the Lithium diffusion activation energy of LiFeSO₄F; 2) computing the lattice parameters and water binding energies for MOF-74; 3) computing the order-disorder transition temperature for Cu-Au alloy; 4) computing the activation energies for Fischer-Tropsch reactions.

This manuscript is well-written and seems to have achieved a significant progress towards creating a universal force field. However, there are several pieces of key information that is missing in the manuscript, making it hard to evaluate the true significance of this work. In fact, the missed information is so important that I believe it should be included for the publication in any scientific journal. I still believe this work could be published at Nature Communications at some point, but the authors need to address the following comments and provide additional information first.

1) Key details regarding the PFP molecule and PFP crystal datasets are missing. The generalization capability of a neural network force field depends on how far the materials in each task are from the closest materials in the training data. For example, in the Lithium diffusion task, the ability to predict activation energy is not supervising if LiFeSO₄F and its randomly perturbed structures are in the training data. The same applies to all four tasks shown in the manuscript.

Therefore, I recommend the authors to add a few sentences in each section to explain what the closest materials are in the training data to the ones in each task. Because GNNs are permutation invariant, the authors also need to explain if any component of material in each task exist in the training data. These statements will help the reader to understand the true generalization capability of the model.

In addition, more details regarding the materials and molecules in both datasets should be included. For example, what is the original source of the crystals and molecules in the PFP datasets? I think the original datasets should be cited and accredited.

2) The details regarding the neural network architecture are missing. Surprisingly, the authors include very few details for the architecture of their NNP model. In their methods section, the authors discussed the invariance, locality, and smoothness of their architecture without providing any details of the actual architecture. This is not acceptable for the publication on a scientific journal

because the readers should be able to reproduce the results claimed by the authors. The authors also did not include a code availability section. From the perspective of the reviewer, the manuscript should not be accepted for publication in any scientific journal if the architecture details are not included.

3) The title is misleading and is clearly an overstatement. I do not believe the authors demonstrated that their model can be applied to an arbitrary combination of 45 elements, because the four tasks are still rather narrow compared with a material from an arbitrary combination of 45 elements. For the title to be appropriate for the manuscript, I believe they need another demonstration to show the model has excellent generalization capability across a broad range of materials. For example, the authors need to demonstrate that the PFP model trained on their PFP molecule and PFP crystal dataset can show acceptable performance on the OCP tasks. Otherwise, I would suggest the authors to choose another title that better reflects their achievements.

4) In the MOF section, the authors mention that “dispersion interaction is also considered and included on top of the computed energies and forces using PFP”. The dispersion interaction calculation defeats the purpose of a neural network potential. The authors should also report the performance of their model without the dispersion correction.

5) It is not very clear to me how the authors tackle the problem of merging DFT datasets with different DFT conditions. The authors select a specific DFT condition to infer during the inference. How would the author choose the condition for a new task?

6) The manuscript lacks empirical results of their model performance. For example, what is the testing energy and force MAEs on their datasets?

Minor comments:

1) Page 16, line 275: NMS is not defined in the manuscript.

2) Page 11, line 215: how did the authors do zero-point energy corrections?

Again, I would like to reiterate that I think this manuscript has made an important step towards a universal neural network potential. However, I do believe that more information needs to be included for the manuscript to be published on Nature Communications.

Reviewer #3 (Remarks to the Author):

The authors present a new approach for machine learning potentials for molecules, materials, and mixtures of the two, with claimed advances both in the neural network architecture and training dataset/approach. The work appears sound, it is clearly explained, and the presented demonstrations are encouraging. Overall, I find the paper very nice, and I believe this work is of broad interest and suitable for Nature Communications if the following points are reasonably considered.

Major comments:

1. The data and code used for training, validating, and testing the models must be documented and provided (to the referees during review, then publicly upon publication). The data/code provided so far (Supp Script 1) is extremely limited.
2. More thorough benchmarking is needed. The three example applications are interesting but pretty limited. For each of these demonstrations, the authors compare to either DFT (Li diffusion) or experiment (MOF, Au-Cu). They should also compare with competitive machine learning models that are already published. This would allow the reader to understand the advantage of the presented approach beyond what is already published. Along a similar vein, there should be more widespread benchmarking presented in this paper. How does the model perform for many hundreds or thousands of materials/molecules/systems that were completely excluded from any aspect of training? Some of this is in the SI in relation to the OC20 dataset, but it is not discussed at all in the manuscript. I feel this is sufficiently important to discuss in the main text of the paper. There should also be similar benchmark demonstrations for molecules and bulk crystals in addition to catalyst surfaces that compare competitive ML approaches, ground-truth (DFT or experiment), and the presented model.
3. There is no discussion of limitations of the model. Do the authors believe the trained model is an adequate replacement for DFT? If so, that claim should be thoroughly justified. If not, the authors

should discuss potential limitations and how they envision the model being used by the materials design community. The EFWT results presented in Table S1 are not overly encouraging to me that the existing model can be an immediate replacement for DFT.

Minor points:

4. Why 45 elements?

5. For the MOF example, where were the initial geometries (before PFP-optimization) taken from?

6. For the Au-Cu example, the transition temperature results are not especially consistent with experiment. Are there other metrics of accuracy that could be considered?

7. A few times, the authors allude to training/testing on disordered structures. I believe this to be a misuse of the term, "disordered". To me, disordered means a material with partial occupancies on one or more site in the structure. In this definition, it is not possible to perform a DFT calculation on a disordered structure, only on an ensemble of fully ordered structures (structures with full or zero occupancies on all sites) that might collectively represent a disordered material. Are the authors' actually training/testing somehow on materials with partial occupancies? The authors should clarify their use of this term.

8. In the Methods, the authors state "we generated structures 305 with randomly placed molecules in addition to the structure-optimized ones using PFP". This implies that structures used for training the PFP were optimized by the PFP. Please clarify this.

Reviewer #1 (Remarks to the Author):

This manuscript reports a machine learning potential trained on both molecular and crystal data with broad coverage across the periodic table, and in principle able to simulate any combination of 45 distinct elements.

This would be a big achievement and definitely, the sort of compelling article that warrants publication. However, the information presented in the paper is very incomplete, to the point that it is hard to evaluate what is actually new (the model? the data?) the benchmarks are all one-off and there is no comparisons with other approaches and the traceability and reproducibility considerations are appalling (No meaningful data availability statements, no code availability statement, no description of the ML model or hyperparameters, no description of the training data or the train/test splits)

There are some language issues (some examples below) that may have hindered my assessment, but my criticism goes well beyond that.

Unfortunately, in this state, a manuscript cannot be considered for publication. The authors would need to conduct a deep revision with a focus on rigorous evaluation of how their model works compared with other baselines, where the innovations and improvements lie, and a lot more focus on what the ML model and the training data are. No one could even attempt to reproduce any of what is reported in this manuscript, with the given information.

Thank you very much for reviewing our manuscript and providing constructive feedback. We have revised the manuscript in response to your comments. We understand that the method section lacks some necessary information. Therefore, we refined the corresponding content. In the dataset section, we provided a detailed description of how each component of the dataset was created in the method based on the reviewers' useful comments. In the architecture section, we focus on the description of our architecture. We believe that these descriptions provide sufficient information for the readers.

Detailed comments:

Data

> "we generate an original dataset that covers all systems" So what exactly is in there? What chemistries? In what numbers? This should be very clear and very up front in the main paper. If the dataset is new, it's a result, not a method, and should be addressed very explicitly and rigorously.

We have modified the Methods section. In addition, we have created a detailed description of each component of the dataset to Supplementary Data 10. We believe that this description is sufficient for readers to reproduce the dataset.

> "we sampled various structures, such as geometrically optimized structures, vibration structures, and MD snapshots," So which was applied to each structure and in which numbers?

We added the correspondence of structure and modification types to Supplementary Data 10.

> "Various structures are generated for a single molecule through geometrical optimization, NMS, and MD at high temperatures." What temperatures? How was NMS done? There is nothing here that allows reproducibility

The settings of the NMS are the same as those in the original method. For the MD simulation, a complex temperature trajectory was used. The calculation flow for the MD simulation is written in the "disordered" section in Supplementary Data 10.

> "we generated structures by changing the cell volumes or shapes, or by randomly displacing the atomic position" By how much? Which random distribution? Uniform? Gaussian? With which parameters?

We added the corresponding description into the "bulk" section in Supplementary Data 10.

> "In addition to the above molecular and crystal datasets, we used the OC20 dataset as a training dataset." Then why is the dataset size smaller than OC20? The whole should be larger than the sum of the parts.

In the dataset comparison table, we only include our dataset for the "PFP molecular dataset (ours)" and "PFP crystal dataset (ours)"

> "The computational resources used to acquire these datasets were approximately 6×10^4 GPU days" what is mean to acquire the datasets? Were they recalculated? Calculated de novo ? What about the CPU time?

This value corresponds to the DFT calculation time, which takes up the majority of the time required to prepare the dataset. The time was aggregated by the built-in tool in the cluster system. The CPU time is not shown here because a major part of the calculation was performed on GPUs in the clusters. In the DFT calculations, CPU costs are considered in managing GPUs.

> "During inference, it is also possible to select which DFT condition to infer by assigning labels in the same way as during training" So which conditions to infer were selected in all the experiments in the paper? Out of which ones?

All applications shown in this study are calculated by the "crystal" calculation mode (GGA-PBE) for which we added a description in the manuscript.

> It is not clear if this paper has innovated a bit, a lot, or nothing at all on the dataset.

We believe that the widely sampled dataset significantly improves the generalizability of NNP. The calculation methods for creating each subcomponent of the dataset and DFT parameters are considered straightforward. As mentioned in the introduction section, the innovative point of this study is to construct a dataset containing unstable structures instead of only collecting known stable

structures, thus enabling the construction of a single NNP model that can describe a diverse set of phenomena.

> Will the dataset be released? Made available upon request? Kept private? The authors seem to praise the open datasets made by others in the introduction (and leverage them in their own work?)

We provide PFP in our software-as-a-service (SaaS) product. Although the exact model parameters and raw dataset are included in the service's intellectual property, and it is not possible to attach these directly to the manuscript, we provided a detailed description of the method for creating each component of the dataset in the Supplementary Information. We believe that these descriptions provide sufficient information for the reader to reconstruct the PFP dataset to apply the method to other tasks, for example, different DFT calculation conditions or different property estimation tasks.

Models

> Table VI is too detailed to be in the main text and an unnecessary distraction. This paper is not comparing against any methods, so why review them? On the other hand, it is not clear what "ours" actually is.

We have modified the description section of the model. We now focus on our model and describe its architecture and configuration. Table VI and corresponding texts are moved into Supplementary Data 8.

> I seem to understand it is the same as TeaNet? Then the authors should be very explicit about if. If it is, the architecture needs to be described layer by layer, interaction block by interaction block. It is not clear if this paper has innovated a bit, a lot, or nothing at all on the interatomic potential part.

Based on the comments, we added a clear explanation on TeaNet which was used for the base architecture of PFP in the Methods section. Since the details of the original TeaNet architecture are beyond the scope of this study, we focused on the description of the modifications of the original model in relation to the objectives of this manuscript. We have also separately updated the original TeaNet manuscript with the step-by-step descriptions of network architecture so that readers can understand the process in-depth and reconstruct it. As written in the modified manuscript, we modified the original TeaNet architecture to adopt the PFP dataset to improve its applicability for users. Although these modifications expand the usage of PFP, they are not considered to have a major impact on the expression power of the architecture.

Experiments:

> Despite all the discussion of dynamics. There are no examples of using the potential to dynamics.

Thank you for your suggestion. Based on the below comment ("Why not test NN-MD for LiFeSO₄F? With such low barrier hopping should be easy to see, at least in one dimension."), we performed MD simulations for LiFeSO₄F and confirmed that the calculated activation energy reproduced the NEB results. We have added the corresponding information to the manuscript and Supplementary Data 13.

> Was LiFeSO4F in the training data? Which geometries for it? The agreement in Table I is outstanding. (DFT itself is barely reproducible to that degree of accuracy and NEB calculations are notoriously fickle to initial parameters or number of images. The agreement is even more surprising due to the difference in the use of U corrections!!)

The crystal structures of LiFeSO4F and FeSO4F are not explicitly included in the dataset. The collected optimized crystal structures are single-element or binary-element systems; therefore, three-body interactions of three different elements cannot be learned from the optimized crystal structure dataset. To the best of our knowledge, the nearest structure is included in a disordered dataset, which contains multiple types of elements together, and there is a chance of having a local configuration similar to that of LiFeSO4F or FeSO4F. In addition, the disordered dataset contains high-temperature structures. We believe that this will enhance the accuracy of non-optimized structures, such as middle replicas of the minimum energy path obtained by the NEB calculation. We have added the corresponding description in Supplementary Data 11.

> "Although neither transition states nor reaction pathways are explicitly given in the training data for creating PFP, " But were MD or NMS or randomly-distorted LiFeSO4F geometries in the training data?"

Similar to the previous reply, there are no structures derived from the LiFeSO4F crystal structure in the dataset.

> Where does the initial structure of LiFeSO4F come from? Was it relaxed with PFP?

The initial structure was obtained from the Materials Project. We optimized the FeSO4F structure directly by removing all Li atoms from the LiFeSO4F structure.

> Why not test NN-MD for LiFeSO4F ? With such low barrier hopping should be easy to see, at least in one dimension.

Please see the reply to the above comment (“Despite all the discussion of dynamics. There are no examples of using the potential to dynamics.”).

> Similar questions arise about the MOF. Are there MOFs in the training data. Is this MOF in the training data. Where does the initial geometry to relax with PFP come from?

MOF structures are not explicitly included in the dataset. To the best of our knowledge, the nearest structure in the dataset is in the disordered dataset with low density.

The experimental crystal structures were obtained from the Cambridge Crystal Structure Database (CCSD). The corresponding CCSD IDs and references are listed in Table S5 in Supplementary Data 14.

> The argument Table II makes is confusing. Often the cell changes less upon hydration than the error w.r.t experiment. Again, how the cells were initialized and whether this a test in the training or in the test domain is of vital importance to evaluate what PFP is bringing to the table.

As Table II is rather convoluted, we have replaced the table with Figure X, showing only the H₂O binding energies and the volumetric difference between the experimentally determined cell geometries and PFP optimized geometries. A table similar to the original Table II is now moved to Supplementary Data 15.

The original crystal structures were cleaned by removing the physically adsorbed solvent molecules from the pores before geometry optimization. The water molecules chemically bound to the open metal centers are kept as they are. For the binding energy calculation, the chemisorbed water molecules were removed, and the geometry was optimized.

This result is considered in the test domain because the MOF structure was not trained explicitly.

> Does Table III recover the rank ? Is Literature experimental or theoretical values?

The literature values are all theoretical DFT calculations. We have replaced Table III with Figure 2 for clarity.

> Does the CuAu experiment use geometrical information? "An arbitrary pair of atoms are swapped, and the energy change (ΔE) is recorded" Were the structures relaxed after the swap? Or kept in the parent lattice parameter?

The structure was optimized after swapping. We have added the description to Supplementary Data 16.

> With so much content (and the need to add so much more) The details of how Metropolis Hastings work can be deferred to the Methods or SI

We have moved the corresponding sentences into Supplementary Data 16.

> How does the reader infer a transition temperature from Figure 3? The agreement with experiment does not seem great and seems to go range around 100 to 200 K. That is worse than a typical cluster expansion

The disordered structures can be observed as diffused points in the figures. We have added a description in the manuscript. We agree that precise domain-specific models can outperform the accuracy. We have rearranged the sentences so that readers can see the numerical errors between our results and the experimental results. In addition, because the Cu-Au disordered structures are not included in the training dataset, this study is also an out-of-domain demonstration of how our PFP works.

> "In this example, we focus our attention on the methanation reactions and

CO dissociation processes on Co surfaces." Was this preset in the training data? Is this from OC20?

Similar to previous applications, we consider this task to be also in the test domain. There is no explicit dataset corresponding to the dissociation of CO molecules. OC20 is a dataset that targets adsorption energy. On the other hand, this time we are targeting the activation energy of dissociation of surface molecules, so we are dealing with a different phenomenon.

Several relevant structures were found in the dataset. OC20 is one of them. In addition, we gathered surface and adsorbed structures. CO molecule structures with various bond distances (corresponding diatomic potential curves) were also included in the dataset. However, the dissociation of CO molecules at the surface structures was not explicitly included in the dataset.

> Were the V experiments confirmed with DFT ?

No. Conducting precise NEB calculations using DFT requires specific modeling to fit the computational cost of DFT, and also incurs large computational costs with trial-and-error. Therefore, we directly compared our results with the experimental data.

> Last but not least. There is no comparison with any baselines whatsoever. Other NN architectures? And/or training over only a datasets?

Thank you for your suggestion. We tested the applications shown in the results section using the OC20 baseline model (DimeNet++). The results have been added to Supplementary Data 12. We confirmed that the OC20 model can reproduce the in-domain case (Fischer-Tropsch catalyst reaction) but cannot reproduce the results of the other out-of-domain cases.

Language:

> "Therefore, PFP possesses the advantages of both universality and a low computational cost" This is clearly not a conclusion of the preceding text

Based on the comment, we removed the sentence from the manuscript.

Reviewer #2 (Remarks to the Author):

In this manuscript, the authors report a universal neural network potential that they claim can be applied to an arbitrary combination of 45 elements. They attempt to solve a key problem with current machine learning based potentials – generalization to new material structures. They created two datasets, PFP molecule and PFP crystal, and developed a GNN based neural network to train a neural network potential. They demonstrate the generalization capability of their model by applying it to 4 very different problems related to material discovery: 1) computing the Lithium diffusion activation energy of LiFeSO₄F; 2) computing the lattice parameters and water binding energies for MOF-74; 3) computing the order-disorder transition temperature for Cu-Au alloy; 4) computing the activation energies for Fischer-Tropsch reactions.

This manuscript is well-written and seems to have achieved a significant progress towards creating a universal force field. However, there are several pieces of key information that is missing in the manuscript, making it hard to evaluate the true significance of this work. In fact, the missed information is so important that I believe it should be included for the publication in any scientific journal. I still believe this work could be published at Nature Communications at some point, but the authors need to address the following comments and provide additional information first.

Thank you very much for reviewing our manuscript and providing constructive feedback. We have revised the manuscript in response to your comments. We addressed the following comments and refine the manuscript.

1) Key details regarding the PFP molecule and PFP crystal datasets are missing. The generalization capability of a neural network force field depends on how far the materials in each task are from the closest materials in the training data. For example, in the Lithium diffusion task, the ability to predict activation energy is not supervising if LiFeSO₄F and its randomly perturbed structures are in the training data. The same applies to all four tasks shown in the manuscript.

Therefore, I recommend the authors to add a few sentences in each section to explain what the closest materials are in the training data to the ones in each task. Because GNNs are permutation invariant, the authors also need to explain if any component of material in each task exist in the training data. These statements will help the reader to understand the true generalization capability of the model.

Thank you for your comment. We have added the corresponding sentences to the Results section. In addition, we have created a detailed description section in Supplementary Data 11.

In addition, more details regarding the materials and molecules in both datasets should be included. For example, what is the original source of the crystals and molecules in the PFP datasets? I think the original datasets should cited and accredited.

We have added the description to Supplementary Data 10. It contains a detailed description of how we created each component of the dataset.

2) The details regarding the neural network architecture are missing. Surprisingly, the authors include very few details for the architecture of their NNP model. In their methods section, the authors discussed the invariance, locality, and smoothness of their architecture without providing any details of the actual architecture. This is not acceptable for the publication on a scientific journal because the readers should be able to reproduce the results claimed by the authors. The authors also did not include a code availability section. From the perspective of the reviewer, the manuscript should not be accepted for publication in any scientific journal if the architecture details are not included.

Based on this comment, we have modified the model description section. It now focuses on our model and describes its architecture and configuration. Table VI and the corresponding texts are moved into Supplementary Information (Data 8). We added an explanation indicating that we used TeaNet for the base architecture of the PFP. As written in the modified manuscript, we modified the original TeaNet architecture to adopt the PFP dataset and improve its applicability for users. We provide PFP in our software-as-a-service (SaaS) product. Although it is not possible to attach the exact implementation and model parameters to the manuscript, The users of the SaaS can use PFP to verify the results written in the manuscript and apply them to other material. In addition, we provided a detailed description of the original TeaNet architecture as a separate manuscript and its modifications in this manuscript. In the core dataset part of PFP, we provided a detailed description of the method for creating each component of the dataset, which can be found in the Supplementary Data 10. We believe that these descriptions provide enough information for the reader to reconstruct the PFP dataset and to apply the knowledge in this study to other datasets and NNP-related tasks.

3) The title is misleading and is clearly an overstatement. I do not believe the authors demonstrated that their model can be applied to an arbitrary combination of 45 elements, because the four tasks are still rather narrow compared with a material from an arbitrary combination of 45 elements. For the title to be appropriate for the manuscript, I believe they need another demonstration to show the model has excellent generation capability across a broad range of materials. For example, the authors need to demonstrate that the PFP model trained on their PFP molecule and PFP crystal dataset can show acceptable performance on the OCP tasks. Otherwise, I would suggest the authors to choose another title that better reflects their achievements.

Thank you for your comment. We modified the title. (“Towards Universal Neural Network Potential for Material Discovery Applicable to Arbitrary Combination of 45 Elements”)

4) In the MOF section, the authors mention that “dispersion interaction is also considered and included on top of the computed energies and forces using PFP”. The dispersion interaction calculation defeats the purpose of a neural network potential. The authors should also report the performance of their model without the dispersion correction.

Thank you for your comment. We have included the results of both PFP and PFP plus the dispersion correction. We originally added the dispersion correction simply because the literature data include the dispersion effect by vdW-DF, and we considered it to be a fair comparison. However, your comment is also reasonable, and we have included both data. The results are summarized in Figure 2. In terms of usage, DFT-D3 dispersion “correction” term is simply added to the DFT calculation result in the original paper of DFT-D3. We took straight-forward approach to only replace DFT calculation

by PFP and add the DFT-D3 dispersion term in this study. We added the corresponding description into the manuscript.

5) It is not very clear to me how the authors tackle the problem of merging DFT datasets with different DFT conditions. The authors select a specific DFT condition to infer during the inference. How would the author choose the condition for a new task?

We have added the description in the Methods section on how to treat multiple DFT conditions in the network architecture. We considered the crystal dataset as the most basic one, and all applications shown in this study are calculated by “crystal” calculation mode (GGA-PBE). We have added a description in the manuscript. Currently, wb97xd is appropriate for a single molecule in a vacuum. We regard the OC20 dataset as supplementary.

6) The manuscript lacks empirical results of their model performance. For example, what is the testing energy and force MAEs on their datasets?

The scatter plot and MAEs of energy and force are presented in Supplementary Data 5. In addition, the calculation time benchmark is provided in Supplementary Data 6.

Minor comments:

1) Page 16, line 275: NMS is not defined in the manuscript.

We added a description of NMS in the manuscript.

2) Page 11, line 215: how did the authors do zero-point energy corrections?

The zero-point energy is calculated using the vibration modes of the PFP. Among all the modes, one mode for the direction of the reaction path with imaginary frequency is excluded.

Again, I would like to reiterate that I think this manuscript has made an important step towards a universal neural network potential. However, I do believe that more information needs to be included for the manuscript to be published on Nature Communications.

Reviewer #3 (Remarks to the Author):

The authors present a new approach for machine learning potentials for molecules, materials, and mixtures of the two, with claimed advances both in the neural network architecture and training dataset/approach. The work appears sound, it is clearly explained, and the presented demonstrations are encouraging. Overall, I find the paper very nice, and I believe this work is of broad interest and suitable for Nature Communications if the following points are reasonably considered.

Thank you very much for reviewing our manuscript and providing constructive feedback. We have revised the manuscript in response to your comments. Please see the replies to the comments written below.

Major comments:

1. The data and code used for training, validating, and testing the models must be documented and provided (to the referees during review, then publicly upon publication). The data/code provided so far (Supp Script 1) is extremely limited.

We modified the method section and now focus on our model and provide a description of its architecture and configurations. We added an explanation that we used TeaNet for the base architecture of the PFP. As written in the modified manuscript, we modified the original TeaNet architecture to adopt the PFP dataset and improve its applicability for users. Separately, we have also updated the original TeaNet manuscript with step-by-step descriptions of network architecture so that readers can acquire an in-depth understanding. We provide PFP in the software-as-a-service (SaaS) product. Although it is not possible to attach the exact implementation and model parameters in the manuscript, The users of the SaaS can use PFP to verify the results written in the manuscript and apply them to other material. In addition, we provided a detailed description of the original TeaNet architecture as a separate manuscript and its modifications in this manuscript. In the core part of PFP, the dataset part, we provide a detailed description of the method for creating the dataset in Supplementary Data 10. We believe that these descriptions provide enough information for the reader to reconstruct the PFP dataset and to apply the knowledge in this study to other datasets and NNP-related tasks.

2. More thorough benchmarking is needed. The three example applications are interesting but pretty limited. For each of these demonstrations, the authors compare to either DFT (Li diffusion) or experiment (MOF, Au-Cu). They should also compare with competitive machine learning models that are already published. This would allow the reader to understand the advantage of the presented approach beyond what is already published. Along a similar vein, there should be more widespread benchmarking presented in this paper. How does the model perform for many hundreds or thousands of materials/molecules/systems that were completely excluded from any aspect of training? Some of this is in the SI in relation to the OC20 dataset, but it is not discussed at all in the manuscript. I feel this is sufficiently important to discuss in the main text of the paper. There should also be similar benchmark demonstrations for molecules and bulk crystals in addition to catalyst surfaces that compare competitive ML approaches, ground-truth (DFT or experiment), and the presented model.

We tested the applications shown in the results section using the OC20 baseline model (DimeNet++). The results have been added to Supplementary Data 12. We confirmed that the OC20 model can reproduce the in-domain case (Fischer-Tropsch catalyst reaction) but cannot reproduce the results of the other out-of-domain cases.

3. There is no discussion of limitations of the model. Do the authors believe the trained model is an adequate replacement for DFT? If so, that claim should be thoroughly justified. If not, the authors should discuss potential limitations and how they envision the model being used by the materials design community. The EFWT results presented in Table S1 are not overly encouraging to me that the existing model can be an immediate replacement for DFT.

We agree that PFP cannot replace the entire DFT calculation immediately. We have added the following sentences to the Discussion section:

“Although DFT calculations or other electronic structure calculations from first principles are still considered to be reliable because of the strong physics background, PFP can greatly mitigate another limitation of atomistic simulations caused by the time and space scales. The combined study of DFT and PFP or experiments using PFP-based screening will also accelerate the field of material discovery.”

Minor points:

4. Why 45 elements?

We have selected from the periodic table the elements which are frequently used in various industrial fields; 45 elements are still on the way and we are still working on expanding the dataset.

5. For the MOF example, where were the initial geometries (before PFP-optimization) taken from?

The starting structures were obtained from the Cambridge Crystal Structure Database (CCSD). The corresponding CCSD IDs and references are listed in Table S5 in Supplementary Data 14.

6. For the Au-Cu example, the transition temperature results are not especially consistent with experiment. Are there other metrics of accuracy that could be considered?

Since we compare the results with experimental data, some phenomena that were not included in the modeling of MD this time may exist. In addition, we agree that precise domain-specific models can outperform the accuracy. We have rearranged the sentences so that readers can clearly see the numerical errors between our results and the experimental results.

7. A few times, the authors allude to training/testing on disordered structures. I believe this to be a misuse of the term, “disordered”. To me, disordered means a material with partial occupancies on one or more site in the structure. In this definition, it is not possible to perform a DFT calculation on a disordered structure, only on an ensemble of fully ordered structures (structures with full or zero

occupancies on all sites) that might collectively represent a disordered material. Are the authors' actually training/testing somehow on materials with partial occupancies? The authors should clarify their use of this term.

We have added a description of the disordered dataset in Supplementary Data 10. We gathered high-temperature structures by MD simulations. We refer to these as disordered structures.

8. In the Methods, the authors state "we generated structures 305 with randomly placed molecules in addition to the structure-optimized ones using PFP". This implies that structures used for training the PFP were optimized by the PFP. Please clarify this.

We have added the corresponding description to Supplementary Data 10 ("disordered" section). We first created an early stage of PFP without MD-based structures and used it to generate MD-based structures.

REVIEWER COMMENTS

Reviewer #1 (Remarks to the Author):

I have reviewed the edits in the authors' revised manuscript and their comments in the rebuttal letter to my and the other reviewer's comments.

In summary, my assessment is that this article reports an intriguing dataset, but is not an intellectually innovative or practically useful contribution to the community.

The results seem to indicate that the dataset is extremely comprehensive, well beyond what has been made available before, and to cover broad composition and structure. The fact that the dataset is not made available, and neither are the trained models (not even the underlying ML architecture is accessible!) makes this contribution little more than useless for the community. There is no intellectual novelty in the ML model (since they are published already as a pre-print but not a peer-reviewed paper and referred to). The dataset may be interesting, but because of its contents, not because of how it was made (which again, there is little of).

For reasons that appear beyond the scientific realm (this seems to be a report of a for-profit product, which is likely in the redacted conflict of interest) the manuscript still lacks comparisons and baselines. The authors have not trained any other models on their new data to compare but rather chosen an obscure ML architecture.

There is now additional detail about the model architecture (TeaNet). This model was reported over two years ago, is not published in a peer-review article, and is not openly accessible. It is not the best choice of model architecture, because it precludes both validation of this work by other parties, and because it has not undergone critical review (but is not the subject of this paper)

There are no comparisons to other architectures trained on the same data. Only to other architectures trained on other data (where TeaNet essentially ties)

The method for generating the dataset is much clearer now, but it is still not reproducible. Every time the paper says "randomly" it is unclear what random distribution and what was the outcome of it. What is the total size of the dataset? How many points for each composition?

If this paper were to receive editorial acceptance despite this reviewer's concern, I would encourage the authors to, at the very least, include a detailed record of what is in this dataset. This document could, for instance, be a csv with one row per entry in the dataset stating:

Composition, atom count, lattice parameters, and "origin" (PFP MD, bulk distortion, etc) along with any metadata for the data generation (random seed, amount of displacement, temperature, etc.)

I am honestly somewhat disappointed in this paper. It could have been a watermark for the community, but the authors seem deliberately obscure in their choices and in their reporting.

As an editorial note, I am in favor of double-blind review policies, but I find the redaction of conflict of interest sections (and funding?) alarming.

Small comments:

Are figures S4 for train, validation or test. What was the criterion for choosing the splits?

Reviewer #2 (Remarks to the Author):

Overall, the manuscript has been significantly improved with more details regarding both the model architecture and the data. I would still recommend the authors consider making the code/data public, but I believe the manuscript is now publishable after minor revisions.

My remaining concern lies in the merging of different DFT modes. The authors simply treat different DFT modes as different labels, and they claim that "it is expected that domains that have only been computed in one DFT condition will be transferred to the inference under other DFT conditions." Can the authors provide additional data to support these statements? For example, what is the test performance on generalizing to different DFT modes? Merging different DFT modes is a challenging problem, and it would be surprising the simple approach employed here can work well.

Reviewer #3 (Remarks to the Author):

The authors have suitably addressed most of the criticisms I raised during the first round of review.

I remain a bit disappointed by the data/code (in)availability. Code and data are withheld as the potential will be provided as a SaaS product. If this adheres to Nature's policy, then I recommend the paper be published.

Before publication, the authors should certainly modify their data availability statement. It states that "Source Data are provided with the paper." This omits the truth. (Limited) data is provided only in the context of the three examples they present. The data used to train/validate the model is NOT available, as far as I can tell.

REPLY TO REVIEWER COMMENTS

Reviewer #1:

> I have reviewed the edits in the authors' revised manuscript and their comments in the rebuttal letter to my and the other reviewer's comments.

> In summary, my assessment is that this article reports an intriguing dataset, but is not an intellectually innovative or practically useful contribution to the community.

> The results seem to indicate that the dataset is extremely comprehensive, well beyond what has been made available before, and to cover broad composition and structure. The fact that the dataset is not made available, and neither are the trained models (not even the underlying ML architecture is accessible!) makes this contribution little more than useless for the community. There is no intellectual novelty in the ML model (since they are published already as a pre-print but not a peer-reviewed paper and referred to). The dataset may be interesting, but because of its contents, not because of how it was made (which again, there is little of).

> For reasons that appear beyond the scientific realm (this seems to be a report of a for-profit product, which is likely in the redacted conflict of interest) the manuscript still lacks comparisons and baselines. The authors have not trained any other models on their new data to compare but rather chosen an obscure ML architecture.

Thank you very much for your review of our paper. We are also very appreciative of your comments. Based on your remarks, we decided to release a portion of our dataset, responding to what we believe to be the most important components. In addition, we included an NNP architecture benchmark using this dataset. The benchmark includes TeaNet, which is our base model, and several popular recently proposed architectures, including NequIP and PaiNN. The hyperparameters of the compared architectures were tuned to this dataset with integrity and as accurately as possible. We confirmed that TeaNet outperforms the benchmark in terms of both energy and force metrics. We believe that both the dataset and architecture benchmark will be informative for readers.

All data related to this benchmark, including the training, validation, and test datasets; architecture implementations; and trained network parameters, are attached to the revised version of the manuscript. Therefore, readers can reproduce the benchmark results, retrain the NNP architectures, and use the dataset for benchmarking the existing or newly proposed NNP architectures.

We believe that these data not only support the main part of our study, they also provide useful knowledge for the relevant academic community and will contribute to future NNP development.

> There is now additional detail about the model architecture (TeaNet). This model was reported over two years ago, is not published in a peer-review article, and is not openly accessible. It is not the best choice of model architecture, because it precludes both validation of this work by other parties, and because it has not undergone critical review (but is not the subject of this paper)

Thank you for your comments. We recently published the corresponding manuscript (TeaNet, <https://arxiv.org/abs/1912.01398>) in a peer-reviewed journal, Computational Materials Science (<https://doi.org/10.1016/j.commatsci.2022.111280>). The article is open access (Creative Commons license, CC BY 4.0), and the corresponding original TeaNet implementation and original dataset are publicly available in Code Ocean (<https://codeocean.com/capsule/4358608>) under widely usable licenses (MIT license for the code and CC0 (no rights reserved) for the dataset). We believe this concern will be addressed based on this progress.

> There are no comparisons to other architectures trained on the same data. Only to other architectures trained on other data (where TeaNet essentially ties)

As described above, the newly attached benchmark contains a comparison between TeaNet and other NNP architectures.

> The method for generating the dataset is much clearer now, but it is still not reproducible. Every time the paper says "randomly" it is unclear what random distribution and what was the outcome of it. What is the total size of the dataset? How many points for each composition?

> If this paper were to receive editorial acceptance despite this reviewer's concern, I would encourage the authors to, at the very least, include a detailed record of what is in this dataset. This document could, for instance, be a csv with one row per entry in the dataset stating:

> Composition, atom count, lattice parameters, and "origin" (PFP MD, bulk distortion, etc) along with any metadata for the data generation (random seed, amount of displacement, temperature, etc.)

Thank you for your comments. We attached the statistical information of all components of our dataset in the Supplementary Information (Supplementary Data 19) section, including the number of structures, number of atoms for each structure, number of elements for each structure, energy

mean and standard deviation, and force mean and standard deviation.

> I am honestly somewhat disappointed in this paper. It could have been a watermark for the community, but the authors seem deliberately obscure in their choices and in their reporting. As an editorial note, I am in favor of double-blind review policies, but I find the redaction of conflict of interest sections (and funding?) alarming.

Again, thank you for your comments. In this study, we decided to make many of the parts of the NNP architecture and dataset public and reproducible. We believe that these data will contribute to community development. We would appreciate your consideration of our manuscript based on this revision.

> Small comments:

> Are figures S4 for train, validation or test. What was the criterion for choosing the splits?

These correspond to the test, and the data points were randomly selected from the corresponding components of the dataset. We added a description of this to the revised version of the manuscript.

Reviewer #2:

> Overall, the manuscript has been significantly improved with more details regarding both the model architecture and the data. I would still recommend the authors consider making the code/data public, but I believe the manuscript is now publishable after minor revisions.

Thank you very much for your review of our manuscript. For this study, we decided to release a part of our dataset and the NNP architecture benchmark. The benchmark includes TeaNet, which is our base model, and several recently proposed architectures. Therefore, readers can reproduce the benchmark and use it for future NNP architecture development. We believe that these data will contribute to the development of a neural network potential community.

> My remaining concern lies in the merging of different DFT modes. The authors simply treat different DFT modes as different labels, and they claim that "it is expected that domains that have only been computed in one DFT condition will be transferred to the inference under other DFT conditions." Can the authors provide additional data to support these statements? For example, what is the test performance on generalizing to different DFT modes? Merging different DFT modes is a challenging problem, and it would be surprising the simple approach employed here can work well.

Based on your suggestion, we added a new simulation result to Supplementary Data 20. We compared the molecular structures of phenol, lithium phenoxide, and sodium phenoxide estimated using both PFP with molecule mode and PFP with crystal mode. We confirmed that the positions of Li and Na atoms, which are not in the molecule mode dataset, can be reproduced based on a molecule mode inference.

Reviewer #3:

> The authors have suitably addressed most of the criticisms I raised during the first round of review.

> I remain a bit disappointed by the data/code (in)availability. Code and data are withheld as the potential will be provided as a SaaS product. If this adheres to Nature's policy, then I recommend the paper be published.

> Before publication, the authors should certainly modify their data availability statement. It states that "Source Data are provided with the paper." This omits the truth. (Limited) data is provided only in the context of the three examples they present. The data used to train/validate the model is NOT available, as far as I can tell.

Thank you very much for your review of our manuscript. For this study, we decided to release a portion of our dataset and the NNP architecture benchmark. The benchmark includes TeaNet, which is our base model, and several recently proposed architectures. Therefore, readers can reproduce the benchmark and use it for the development of future NNP architectures. Based on the above modification, we modified the data and code availability statements. We believe that these modifications will resolve this issue.

REVIEWERS' COMMENTS

Reviewer #1 (Remarks to the Author):

The authors have successfully addressed the bulk of my criticisms. With comparisons between models trained on the same data and a benchmark dataset, the manuscript is finally a worthy and helpful contribution to the field.